# Risk Zoning of Typhoon Disasters in Zhejiang Province, China

Yi Lu[1]（陆逸）, Fumin Ren[2]（任福民）, Weijun Zhu[3]（朱伟军）

[1] Shanghai Typhoon Institute of China Meteorological Administration,Shanghai 200030, China

[2] State Key Laboratory of severe weather; Chinese Academy of Meteorological Sciences,Beijing 100081, China

[3] Key Laboratory of Meteorological Disaster of Ministry of Education, Nanjing University of Information

Science &Technology, Nanjing 210044, China

**Abstract** In this paper, typhoon simply means tropical cyclone. As risk is future probability of hazard events, when suppose future probability is the same as historical probability for a specific period, we can understand risk by learning from past events. Based on precipitation and wind data over the mainland of China during 1980 - 2014 and disaster and social data at the county level in Zhejiang province from 2004 to 2012, a study on risk zoning of typhoon disasters (typhoon disasters in this paper refer to affected population or direct economic losses caused by typhoons in Zhejiang province) is carried out. Firstly, characteristics of typhoon disasters and factors causing typhoon disasters are analyzed. Secondly, an intensity index of factors causing typhoon disasters and a population vulnerability index are developed. Thirdly, combining the two indexes, a comprehensive risk index for typhoon disasters is obtained and used to zone areas of risk. The above analyses show that, southeastern Zhejiang is the area most affected by typhoon disasters. The annual probability of the occurrence of typhoon rainstorms >50 mm decreases from the southeast coast to inland areas, with a maximum in the boundary region between Fujian and Zhejiang, which has the highest risk of rainstorms. Southeastern Zhejiang and the boundary region between Zhejiang and Fujian province and the Hangzhou Bay area are most frequently affected by typhoon extreme winds and have the highest risk of wind damage. The population of southwestern Zhejiang is the most vulnerable to typhoons as a result of the relatively undeveloped economy, mountainous terrain and the high risk of geological disasters in this region. Vulnerability is lower in the cities due to better disaster prevention and reduction strategies and a more highly educated population. The southeast coastal areas face the highest risk of typhoon disasters, especially in the boundary region between Taizhou and Wenzhou cities. Although the inland mountainous areas are not directly affected by typhoons, they are in the

Corresponding author: Dr. Fumin Ren, State Key Laboratory of severe weather (LaSW)/CAMS, Beijing, 100081.
E-mail address: fmren@163.com

medium-risk category for vulnerability.
**Keywords:** typhoon disasters, factors causing typhoon disasters, vulnerability, comprehensive risk
index, risk zoning

## 1 Introduction

Typhoon, which means tropical cyclone in this paper, often causes some of the most serious natural
disasters in China, with an average annual direct economic loss of about $9 billion. The arrival of
typhoon is often accompanied by heavy rain, high winds and storm surges, with the main impacts in
southern coastal areas of China (Zhang et al., 2009). Zhejiang province is seriously affected by
typhoons—for example, in 2006, super-typhoon Sang Mei caused 153 deaths in Cangnan county of
Wenzhou city, with 11.25 billion yuan of direct economic losses. Therefore it would be of practical
significance to develop a system for the risk assessment of typhoon disasters in Zhejiang province.
Major risk assessment models include the disaster risk index system of the United Nations
Development Program (global scale, focusing on human vulnerability), the European multiple risk
assessment (with emphasis on factors causing disasters and vulnerability) and the American
HAZUS-MH hurricane module and disaster risk management system. Vickery et al. (2009) and Fang et
al. (2012, 2013) reviewed the factors causing typhoon disasters. Rain and wind are direct causes of
typhoon disasters (Emanuel, 1988, 1992, 1995; Holland, 1997; Kunreuther and Roth, 1998); stronger
typhoons produce heavier rain and stronger winds, resulting in a greater number of casualties and
higher economic losses. Many of the researches on the factors causing typhoon disasters used a grade
index and the probability of occurrence (Chen et al., 2011; Su et al., 2008; Ding et al., 2002; Chen,
2007). Recently, some research built quantitative assessment in some provinces and carried out
preliminary studies on pre-evaluating typhoon disasters (Huang and Wang, 2015; Yin and Li, 2017).
In terms of vulnerability, Pielke et al. (1998, 2008) combined the characteristics of typhoons and
socioeconomic factors, suggesting that both the vulnerability of the population and economic factors
were important in estimating disaster losses. The vulnerability of a population is a pre-existing
condition that influences its ability to face typhoon disasters. Among the most widely used indexes is
the Social Vulnerability Index (SoVI) (Cutter et al., 2003; Chen et al., 2011). Other researches have
focused on the vulnerability of buildings, obtaining a fragility curve by combining historical loss with
the characteristics of buildings and typhoons (Hendrick and Friedman, 1966; Howard et al., 1972;
Friedman, 1984; Kafali and Jain, 2009; Pita et al., 2014). Studies in China have assessed vulnerabilities
to typhoon disasters (Yin et al., 2010; Niu et al., 2011). Evaluation indexes for the assessment of
disaster losses were established based on the number of deaths, direct economic losses, the area of
crops affected and the number of collapsed houses. These indexes were used to construct different
disaster assessment models (Liang and Fan, 1999; Lei et al., 2009; Wang et al., 2010). Xu et al. (2015)
comprehensively assessed the impact of typhoons across China using the geographical information
system. The future direction of tropical cyclone risk management is quantitative risk models (Chen et
al., 2017).
Previous studies have concentrated on semi-quantitative, large-scale research, with less emphasis
on quantitative research at county level based on large amounts of accurate data. In addition, the studies
have paid more attention to disaster losses. Few studies have focused on a comprehensive risk
assessment of typhoon disasters coupled with factors causing typhoon disasters and population
vulnerability. In this study, Zhejiang province, which is frequently affected by the strongest landfall
typhoons (Ren et al., 2008) and experiences most serious typhoon disasters (Liu and Gu, 2002) in the
mainland of China, is selected as the study area. This paper does not consider the impact of storm
surges. The factors causing typhoon disasters are represented by typhoon rain and typhoon wind.
Section 2 introduces the data and methods used in this study. Section 3 provides analyses on typhoon
disaster losses and causing factors. Section 4 presents risk assessment and regionalization of typhoon
disasters. Summary and discussions are given in the final section.
**2 Data and Methods**
This study is carried out in Zhejiang province (Figure 1) including 11 cities along the Yangtze River
Delta. Zhejiang province is in the eastern part of the East China Sea and north to Fujian province,
which is one of the most economically powerful provinces in China.

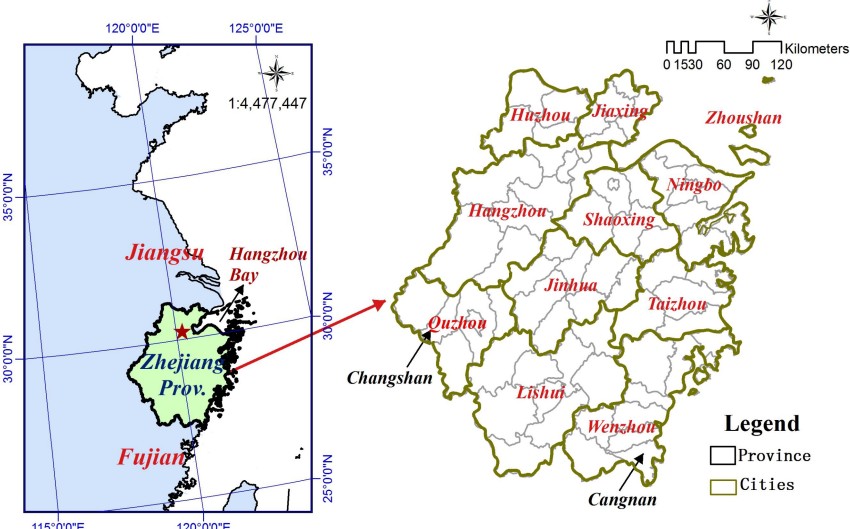

Figure 1. Maps of Zhejiang province, China showing location and major cities.
**2.1 Data**
**2.1.1 Typhoon, Precipitation and Wind Data**
The typhoon data used in this study are the best-track tropical cyclone datasets from Shanghai Typhoon
Institute for the time period 1960 - 2014 (Eunjeong and Ying, 2009; Li and Hong, 2015). Daily
precipitation data for 2479 stations and daily wind data for 2419 stations during the time period 1960 -
2014 over the mainland of China are obtained from the National Meteorological Information Center.
The maximum wind speed is given as the maximum of 10-minute mean. In this paper, two time periods
of precipitation and wind data are used.
Because of limited access to county-level typhoon disaster data, we have only obtained data
during 2004 to 2012. So when calculating intensity index of factors causing typhoon disasters, the time
period of typhoon precipitation and typhoon wind are the same as typhoon disasters, which is 2004 -

95    2012.

For risk analyses of typhoon precipitation and typhoon wind (please see detail in sections 3.1 and
3.2), suppose future probability is the same as historical probability, we then select the period of 1980 –
2014. As Lu et al. (2016) mentioned, considering the homogeneity of wind data, we use the period of
1980 - 2014 for wind analysis. To ensure the consistency between wind and precipitation data, 1980 -
2014 is selected as the period. In addition, the Objective Synoptic Analysis Technique (OSAT) method
need to identify typhoon wind and precipitation from a wider range than Zhejiang province (please see
details in section 2.2.1), so 2419 stations of precipitation data and 2479 stations of wind data over the
mainland of China are used, which all contained 71 stations corresponding to counties in Zhejiang
province.
**2.1.2 Disaster and Social Data**
Disaster data for each typhoon that affected Zhejiang province from 2004 to 2012 are obtained from the
National Climate Center and the number of records for each county is shown in Figure 2. Of the 11
cities in Zhejiang province, Wenzhou and Taizhou record the most typhoon disasters, with a maximum
being 17 at Wenzhou. Fewer typhoon disasters are recorded in the central and western regions of
Zhejiang province, particularly in Changshan and Quzhou, which may be because the strength of
typhoons weakened after landfall. The population data in 2010 are obtained from the sixth national
population census (Population Census Office of the National Bureau of Statistics of China), and the
2010 statistical yearbooks of each city in Zhejiang province published by the cities' statistical bureaus.
The census data is updated every six years, and the 2010 census results are exactly during 2004-2012
which is the research period. Therefore, the population data for 2010 in this paper can basically
represent the population vulnerability of this period. Basic geographical data are obtained from the
National Geomatics Center of China.

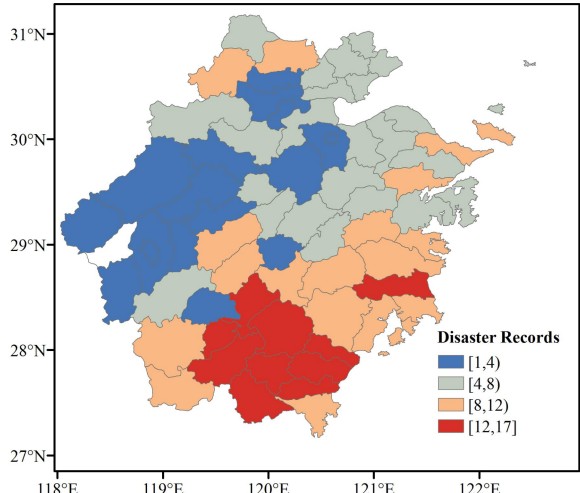


119            Figure 2. Number of records of typhoon disasters in Zhejiang province from 2004 to 2012.

**2.2 Methods**
**2.2.1 Objective Synoptic Analysis Technique**

The widely used objective synoptic analysis technique (OSAT) proposed by Ren et al. (2001, 2007, 2011) is used to identify precipitation due to typhoons in this study. The OSAT method is a numerical technique to separate tropical cyclone induced precipitation from adjacent precipitation areas. Based on structural analysis of precipitation field, it can be divided into different rain belts. Then, according to the distances between a TC center and these rain belts, typhoon center and each station, typhoon precipitation is distinguished. Lu et al. (2016) improved the OSAT method and applied it to identify typhoon winds. With the application of the OSAT method, daily precipitation and wind data over the mainland of China during 1980 to 2014 are used for identifying typhoon precipitation and wind data.

**2.2.2 Canonical Correlation Analysis (CCA)**

We use the canonical correlation analysis method to determine the relationship between the affected population, the rate of economic damage, and typhoon precipitation and winds. In statistics, canonical correlation analysis (CCA) is a way of inferring information from cross-covariance matrices. If we have two vectors $X = (X_1, ..., X_n)$ and $Y = (Y_1, ..., Y_m)$ of random variables, and there are correlations among the variables, then CCA can find linear combinations of the $X_i$ and $Y_j$ which have maximum correlation with each other (Hardoon et al., 2014). The method was first introduced by Hotelling in 1936 (Hotelling, 1936). The main point of CCA is to separate linear combinations of new variables from the two sets of variables. In this case, the correlation coefficient between new variables reaches the maximum. In this paper, we chose factors causing typhoon disasters as a set of variables, and typhoon disaster as another. Under the maximum canonical correlation coefficient, the linear combination coefficients (typical variable coefficients) of factors causing typhoon disasters can be used as weight coefficients of this group of variables. Then we can determine the impact of factors causing typhoon disasters.

**2.2.3 Data Standardization**

We adopt two methods: Z-score standardization and MIN-MAX standardization. The Z-score standardized method is based on the mean and standard deviation of the raw data, which is prepared for CCA method. The MIN-MAX standardization is a linear transformation of the original data so that the original value maps the interval [0, 1]. Z-score standardization is used for calculating the intensity index of factors causing typhoon disasters. Both typhoon precipitation and typhoon maximum wind speed are standardized by this method. When calculating the typhoon disaster comprehensive risk

index (R), we use MIN-MAX standardization to standardize the intensity index of the factors causing
typhoon disasters (I) and the population vulnerability index (SoVI).
**2.2.4 Vulnerability Assessment (SoVI, PCA)**
County-level socioeconomic and demographic data are used to construct an index of social
vulnerability to environmental hazards named the Social Vulnerability Index (SoVI). Principal
Component Analysis (PCA) is the primary statistical technique for constructing the SoVI. The PCA
method captures multi-dimensionality by transforming the raw dataset to a new set of independent
variables. Then a few components can represent the dimensional data, and underlying factors can be
identified easily. These new factors are placed in an additive model to compute a summary
score—SoVI (Cutter et al., 2003). Based on various SoVIs derived for disaster social vulnerability in
America, Chen et al. (2014) collects 29 variables as proxies to build a set of vulnerability indexes for
the social and economic environment in China. We then use these vulnerability indexes to calculate the
population vulnerability index for Zhejiang province.
**3 Typhoon Disaster Losses and Factors**
Based on the distribution of typhoon disaster losses in Zhejiang province from 2004 to 2012 (Figure 3),
the affected areas are mainly locates in the southeast corner of the province. The centers with the
largest affected population (Fig. 3a), the largest area of affected crops (Fig. 3c) and the highest direct
economic losses (Fig. 3d) are in Wenzhou and Taizhou cities, although the losses in Ningbo City are
also relatively high. Cangnan in Wenzhou City is the most severely affected, with the highest
cumulative death toll (Fig. 3b). According to the statistical yearbooks of each city in Zhejiang province,
Jiaxing, Shaoxing, Hangzhou in the northeast, and Wenzhou, Jinhua and Taizhou in the southwest are
the regions with the largest agricultural planting area, with more agricultural population in the
southwest. Only parts of the plain area were affected by serious agricultural disasters in the northeast.
The agricultural disaster areas in the southwest are wider. (Fig. 3c). According to the main indicators of
Zhejiang's national economy (total GDP and per capita GDP), the central cities such as Hangzhou in
the northeast had the most developed economy, and the urban economies of Wenzhou and Taizhou in
the southwest were also relatively good. However, the economic losses in southwestern Zhejiang are
severe, much higher than in the northeastern cities. (Fig. 3d). The losses in the affected counties are
associated with the frequency and intensity of typhoons. We therefore analyze the risk of typhoon
precipitation and winds in every county in Zhejiang province to provide a reference dataset for the
factors causing typhoon disasters.

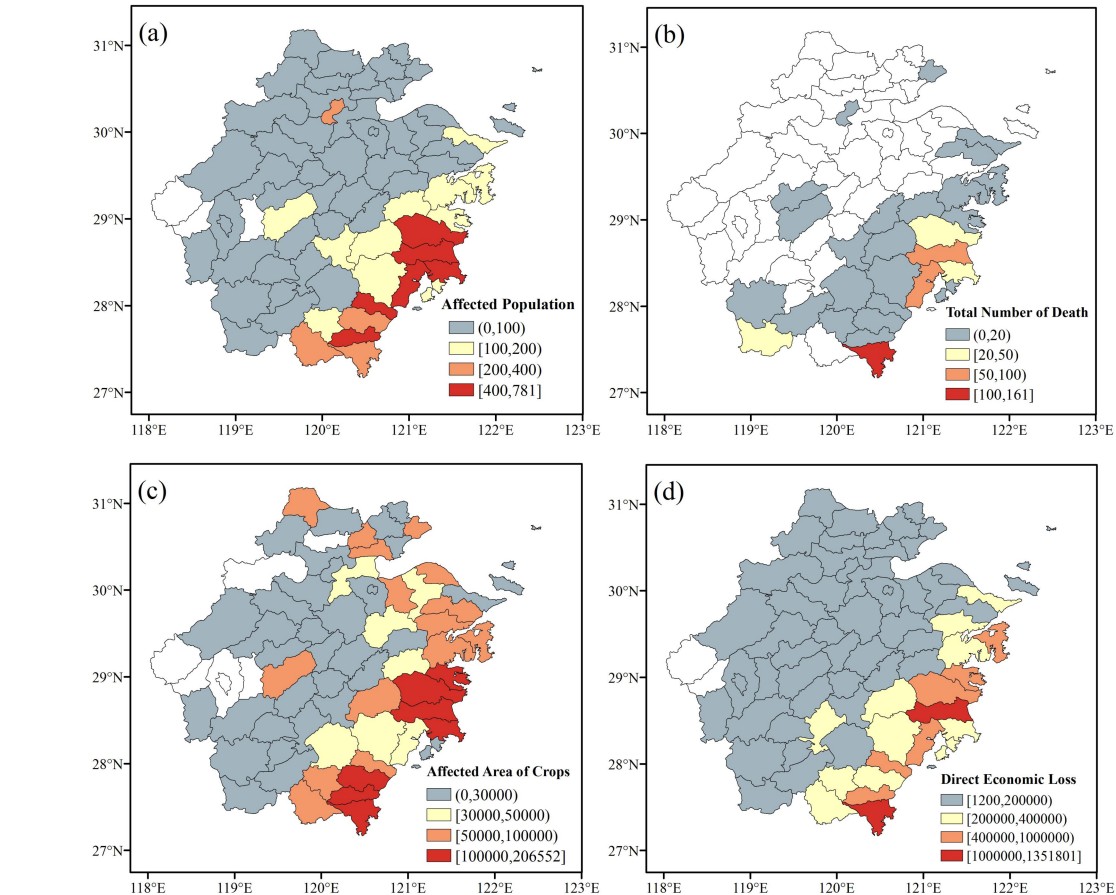

Figure 3. Distribution of typhoon disaster losses in Zhejiang province from 2004 to 2012. (a) Affected
population (millions); (b) total number of deaths (person); (c) area of affected crops (hectares); and (d)

186               direct economic losses (millions yuan).

**3.1 Probability of Typhoon Rainstorms**
The main hazard of typhoon precipitation is concentrated precipitation, so the average duration (days)
of typhoon precipitation at each station in Zhejiang province is counted from 1980 to 2014 (Figure 4).
The duration of typhoon rainfall is less in inland areas, especially in Quzhou City. Persistent
precipitation is concentrated in Wenzhou, Taizhou and Ningbo cities, where there may have been a
higher risk of typhoon disasters. Typhoon rainstorm in this study means daily typhoon precipitation
over 50mm, and typhoon torrential rainstorm means daily typhoon precipitation over 100mm. The
probability is the annual possibility of the occurrence of typhoon rainstorms. The probability
denominator is the total number of years, and the numerator is the annual frequency of typhoon

precipitation. If a station experiences typhoon precipitation in one year, the numerator increases by one. Based on the probability of typhoon rainstorms occurring in each county in Zhejiang province (Figure 5), we found that the annual probability of the occurrence of typhoon rainstorms is highest over the southeast coast of Zhejiang province from 1980 to 2014, especially in Taizhou City, where the annual probability is 83%. The annual probability of typhoon rainstorms with precipitation >100 mm is lower, but the distribution of probability is consistent with the rainstorms with lower precipitation. The probability of typhoon torrential rainstorms decreases rapidly in the western and central regions of Zhejiang province, although the range increases. There are three centers of high probability: Taizhou, Wenzhou and Ningbo cities.

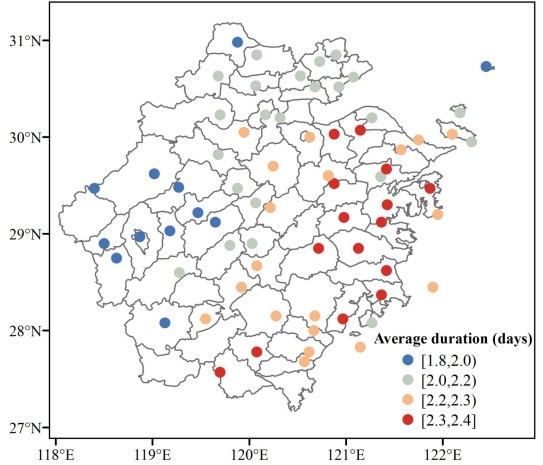

Figure 4. Average duration (days) of typhoon precipitation at each station in Zhejiang province from 1980 to 2014.

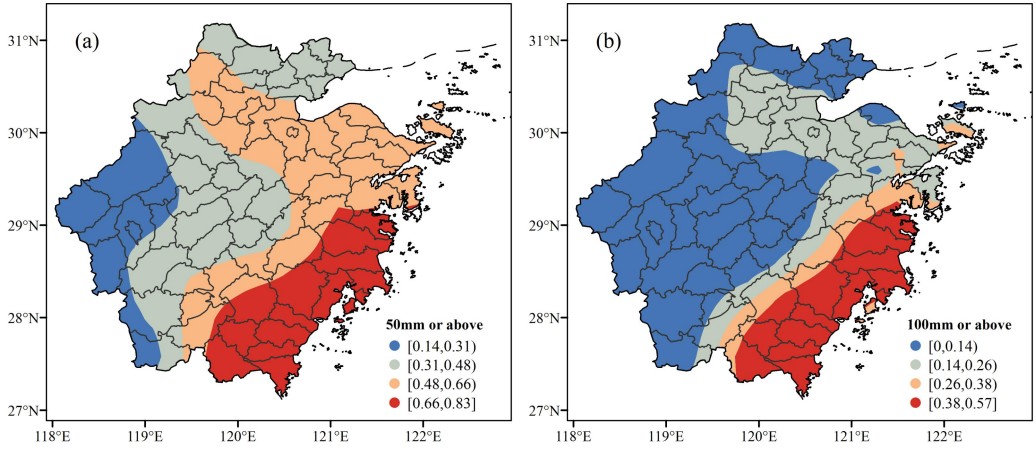

Figure 5. Probability of the occurrence of typhoon rainstorms in Zhejiang province: (a) rainstorms with precipitation >50 mm; and (b) torrential rainstorms with precipitation >100 mm.

**3.2 Probability of Typhoon Winds**
The average duration (days) of typhoon winds (over 6 grade) is calculated in Zhejiang province (Figure
6). The duration of typhoon winds is relatively short in the central and western regions and the typhoon
winds are concentrated in the coastal areas of Wenzhou, Taizhou and Ningbo cities. The longest
duration of typhoon winds occurs over the offshore islands.
The main hazard from typhoon winds is manifested in the destructive force of strong winds.
Therefore we calculate the probability of annual occurrence of typhoon winds at or above grades 6 and
12 at each station from 1980 to 2014 (Figure 7). Typhoon winds at or above grade 6 mainly occur along
the coastal areas, with rare occurrence in the mountainous areas. Meanwhile, the probability of typhoon
winds at or above grade 8 is generally 0.5~0.9 along the coast, and below 0.25 in the inland
mountainous areas. Typhoons winds at or above grade 10 or 12 are much less likely and are only seen
in the coastal areas and islands, with a rapidly decreasing probability from the coastal areas to the
inland mountainous areas. The areas at high probability of typhoon winds are consistent with those
with a high probability of typhoon rain, i.e. Wenzhou, Taizhou and Ningbo cities. The probability of
typhoon extreme winds is much higher in coastal areas than inland.

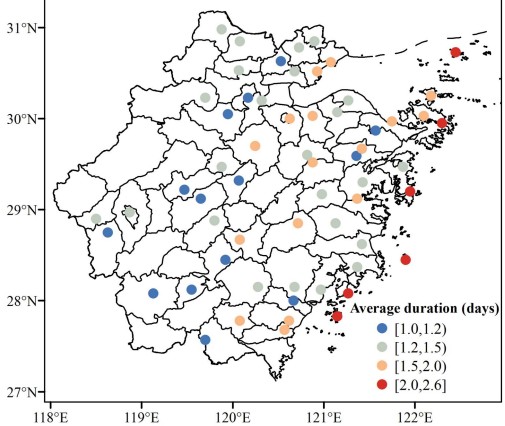


Figure 6. Average duration (days) of typhoon winds (over 6 grade) at each station in Zhejiang province
from 1980 to 2014.

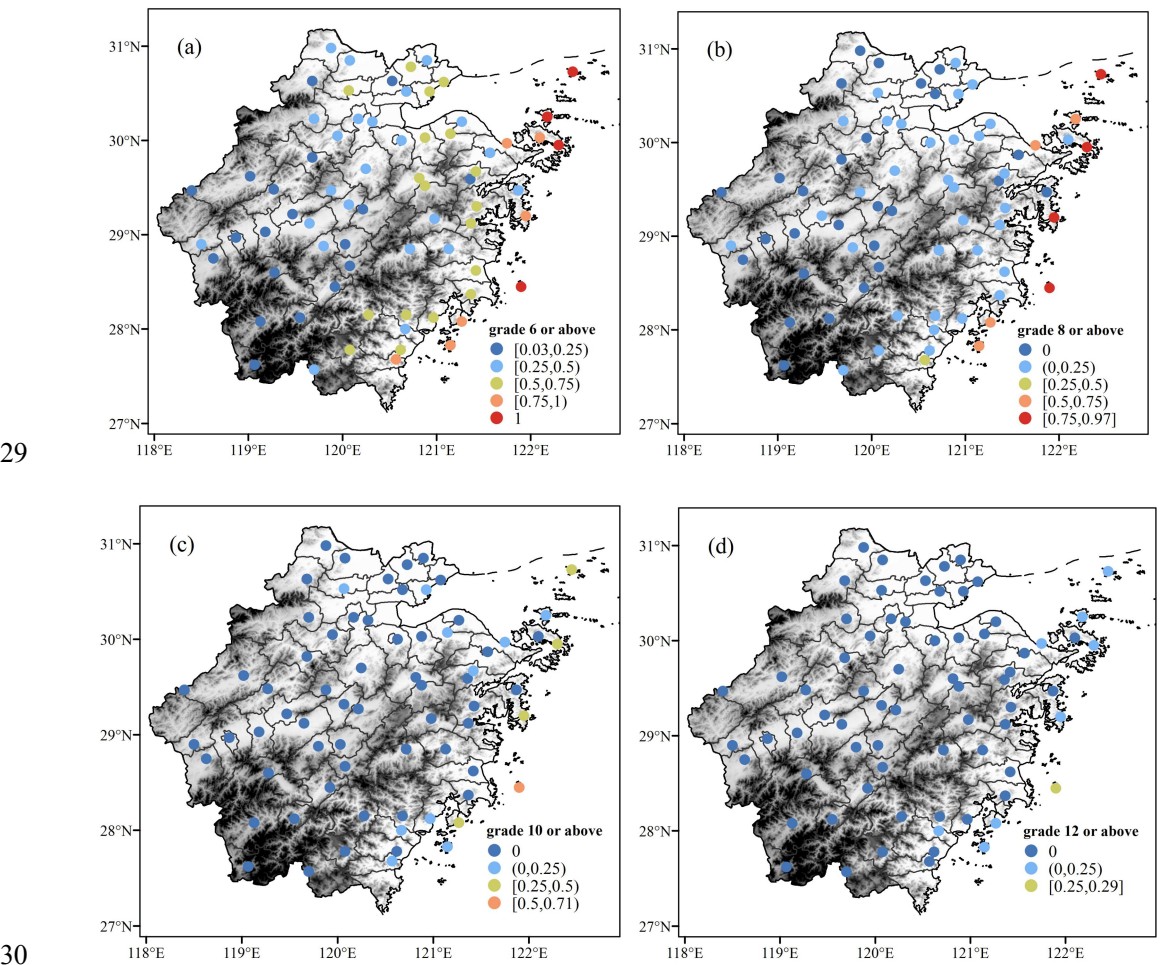



Figure 7. Probability of the occurrence of typhoon winds in Zhejiang province at (a) grade 6 or above

(≥ 10.8 m/s), 672 (b) grade 8 or above (≥ 24.5 m/s), (c) grade 10 or above (≥ 32.7 m/s) and (d) grade 12

or above (≥ 41.5 m/s) .

## 4 Risk Assessment and Regionalization of Typhoon Disasters

### 4.1 Intensity Index of Factors Causing Typhoon Disasters

The main factors causing typhoon disasters, which are considered in this study, are rainstorms and winds. The level and intensity of a single factor cannot fully represent and describe the impact. It is necessary to determine their influence through typical correlation analysis, and then typhoon wind and rain effect are superimposed by the weight coefficients. Therefore we establish a comprehensive intensity index that includes typhoon precipitation and winds. Taking the county as a unit, we select all the typhoons that affected the population of Zhejiang province from 2004 to 2012. The total

precipitation and daily maximum wind speed during typhoons measured in each county are used to
describe the factors causing typhoon disasters. The total sample size is 322. Using CCA, we determine
the impact of typhoon precipitation and winds on the population. We then do CCA for all the typhoons
that caused direct economic losses in Zhejiang province from 2004 to 2012, and the total sample size is
404 (Table 1). The effect of typhoon precipitation on both the population and direct economic losses is
always greater than that of typhoon winds. By averaging typical coefficients for both precipitation and
wind, weight coefficients of 0.85 and 0.65 are obtained within the intensity index for precipitation and
winds, respectively.
Table 1. Canonical correlation analysis of factors causing typhoon disasters.

| Disasters | Canonical correlation coefficient | Canonical variable coefficient | |
| --- | --- | --- | --- |
| | | Typhoon precipitation | Typhoon wind |
| Affected population | 0.45 | 0.84 | 0.651 |
| Direct economic losses | 0.477 | 0.863 | 0.655 |


Based on the weight coefficients in Table 1, an intensity index of factors causing typhoon
disasters is established:
$I = Ax + By$        (1)
where $I$ is the intensity index of factors causing typhoon disasters, $X$ is the standard typhoon
precipitation and $Y$ is the maximum wind speed of the typhoon. $A$ and $B$ are the weight coefficients for
typhoon precipitation and typhoon winds, respectively. Using Equation (1), we average the intensity
indexes of typhoons at each station (Figure 8). Based on the distribution of these average intensity
indexes, three high value centers, namely Wenzhou, Taizhou and Ningbo cities are identified, which is
consistent with the results of Chen et al. (2011), can be found.

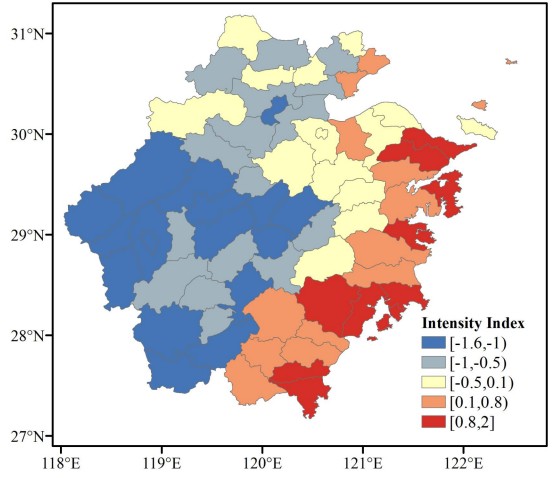


Figure 8. Intensity indexes of factors causing typhoon disasters at each station in Zhejiang province.
**4.2 Population Vulnerability Index**
Natural disasters are social constructions and the basic causes of losses are the attributes of human
beings and their social system (Jiang 2014). The index system of Chen et al. (2011) is used to evaluate
the vulnerability of Zhejiang province. Based on the extracted population information, 29 variables are
identified that may affect vulnerability (Table 2).
Table 2. The 29 variables affecting vulnerability in Zhejiang province.

|  | variables | Name |
|---|---|---|
| 1 | Per capita disposable income of urban residents (yuan) | UBINCM |
| 2 | Percentage of female (%) | QFEMALE |
| 3 | Percentage of minority (%) | QMINOR |
| 4 | Median age | MEDAGE |
| 5 | Unemployment rate (calculated - unemployed population / (unemployed + total population) | QUNEMP |
| 6 | Population density | POPDEN |
| 7 | Percentage of urban population (%) | QUBRESD |
| 8 | Percentage of non-agricultural household population (%) | QNONAGRI |
| 9 | Percentage of households that living in rented houses (%) | QRENT |
| 10 | Percentage of employees working in primary industries and mining (%) | QAGREMP |
| 11 | Percentage of employees working in secondary industries (%) | QMANFEMP |
| 12 | Percentage of employees working in tertiary industries (%) | QSEVEMP |

| | | |
|---|---|---|
| 13 | Household size (person / household) | PPUNIT |
| 14 | Percentage of population with college degree (25 years old and older) | QCOLLEGE |
| 15 | Percentage of population with high school degree (20 years old and older) | QHISCH |
| 16 | Percentage of illiterate people (15 years old and older) | QILLIT |
| 17 | Population growth rate (2000-2010) | POPCH |
| 18 | Average number of rooms per household (inter / household) | PHROOM |
| 19 | Per capita housing construction area ($m^2$ / person) | PPHAREA |
| 20 | Percentage of premises without tap water (%) | QNOPIPWT |
| 21 | Percentage of premises without a kitchen (%) | QNOKITCH |
| 22 | Percentage of premises without a toilet (%) | QNOTOILET |
| 23 | Percentage of premises without a bath (%) | QNOBATH |
| 24 | Number of beds per 1000 person in health care institutions | HPBED |
| 25 | Number of medical personnel per 1000 resident population | MEDPROF |
| 26 | Percentage of people under 5 | QPOPUD5 |
| 27 | Percentage of population over 65 years old | QPOPAB65 |
| 28 | Population dependency ratio (%) | QDEPEND |
| 29 | Percentage of population covered by subsistence allowances (%) | QSUBSIST |

After Principal Component Analysis (PCA) of the 29 variables, seven components with
eigenvalue >1 are extracted. Based on the variable meanings in each component, these 7 components
are named as table 3. The first component, which reflects the income of the population and the
employment situation, contribute 30.1% of the total variance. This component is positive because the
more property there is in an area, the higher the vulnerability to damage. The second component, which
reflects education level of the population, occupies 15.6% of the total variance. This component is
negative because if education level is higher, then the population's awareness of disaster prevention and
reduction is greater and their vulnerability is lower. The third component, which reflects the number of
dilapidated houses, takes up 8.7% of the total variance. This component plays a positive part in
vulnerability. The fourth component, which reflects the illiteracy and the number of young people, is
positive and represents 8.4% of the total variance. The fifth component, which reflects the household
size and the percentage of women, explains 7.7% of the total variance and is positive. The sixth
component, which reflects the number of ethnic minorities, contributes 6.1% of the total variance and
is positive. The seventh component, which represents 5.3% of the total variance, reflects the
unemployment rate and the housing area and is positive.

285        The total variance explained by these seven components is up to 81.9%, which can be used to

represent the population vulnerability of Zhejiang province. The distributions of the first (positive)
component and the second (negative) component are shown in Figure 9. Areas with a low employment
rate have high vulnerability, but the vulnerability is low in urban areas with higher levels of education.
The seven components thus represent the real situation of the population vulnerability in Zhejiang
province to the effect of typhoons. The population vulnerability index in Zhejiang province (SoVI) is
calculated as:

292        SoVI= component 1 − component2 + component 3 + component 4 + component 5 + component

6 + component 7                                                                              (2)

294        By calculating the vulnerability indexes of each county, the distribution of population

vulnerability in Zhejiang province is obtained (Figure 10). The areas with high vulnerabilities are
mountainous regions where the economy is relatively undeveloped, whereas the vulnerability is low in
cities, such as Hangzhou and Huzhou cities, where there is a greater awareness of disaster prevention
and reduction and houses are of high quality.

299                        Table 3. The seven components extracted by PCA.

| Components | Contained variables | Name | (Sign) |
|---|---|---|---|
| 1 | QMANFEMP, UBINCM, QAGREMP, QRENT, POPCH, QDEPEND, QSUBSIST, QPOPAB65, POPDEN, MEDAGE, QNOKITCH, QILLIT, PHROOM, PPHAREA | Employment and poverty | (+) |
| 2 | QHISCH, QCOLLEGE, QNONAGRI, QSEVEMP, HPBED, MEDTECH | Education | (-) |
| 3 | QNOBATH, QNOTOILET, PPUNIT | Number of dilapidated houses | (+) |

| | | | |
|---|---|---|---|
| 4 | QILLIT, QDEPEND, QPOPUD5, MEDAGE | Illiteracy and juvenile population | (+) |
| 5 | QFEMALE, PHROOM, PPHAREA, QSEVEMP | Household size and ratio of women | (+) |
| 6 | QMINOR | Ethnic minority | (+) |
| 7 | QUNEMP, QNOPIPWT | Unemployment and housing size | (+) |


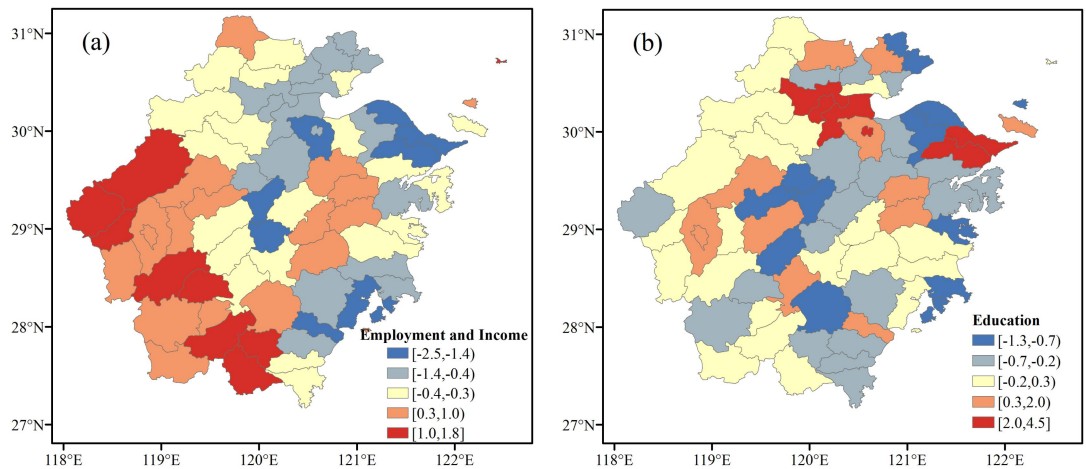


Figure 9. Distribution of population vulnerability index of (a) component 1 (employment and
income) and (b) component 2 (education).

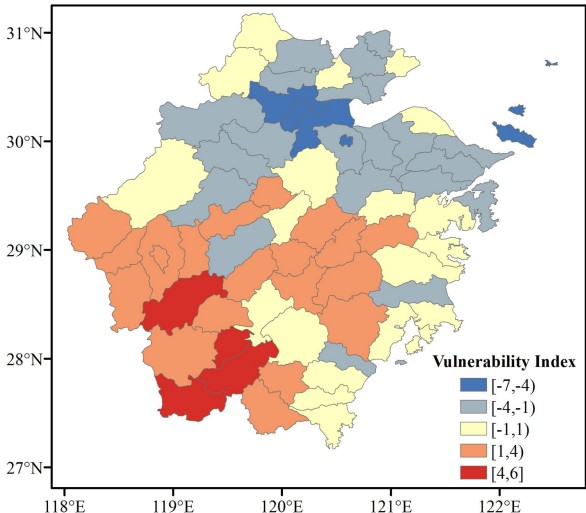

Figure 10. Distribution of population vulnerability index of counties.

**4.3 Typhoon Disaster Comprehensive Risk Index and Zoning**

The typhoon disaster risk assessment system is mainly composed of the factors causing disasters, the population vulnerability and the environment. In this paper, typhoon disaster comprehensive risk index is obtained by combining the factors causing typhoon disasters and vulnerability, without taking the sensitivity of the environment into account. After standardizing the intensity index of factors causing typhoon disasters and the population vulnerability index, the typhoon disaster comprehensive risk index (*R*) is obtained as follows:

$R$ = intensity index of factors causing typhoon disasters (*I*) ×vulnerability index (SoVI)     (3)

Based on the comprehensive risk index, five risk grades for typhoon disasters are defined (Table 4) , and risk zoning of typhoon disasters in Zhejiang province has been done as shown in Figure 11. The classification of typhoon disaster risk index is based on the natural breaks method (Jenks) provided by Arcgis.

Table 4. Disaster risk index and grading.

| Risk grade: | High | High–medium | Medium | Medium–low | Low |
|---|---|---|---|---|---|
| Risk index: | 0.3 | 0.18–0.3 | 0.13–0.18 | 0.07–0.13 | 0.07 |

Figure 11 shows that, the index presents a good reflection of the distribution of typhoon disasters in Zhejiang province (Figure 3), especially in the southeastern coastal areas. The southeast coastal areas face the highest risk, especially in the boundary regions between Zhejiang and Fujian province, and

Taizhou and Wenzhou cities. Overall, the risk of typhoon disasters decreases from the coast to inland
areas. Cities are at medium to low risk as a result of their developed economy, high-quality houses and
better educated population. The inland mountainous areas have a high vulnerability. Although they are
not directly affected by typhoons, they are still in the middle risk areas as a result of their poorly
developed economy.

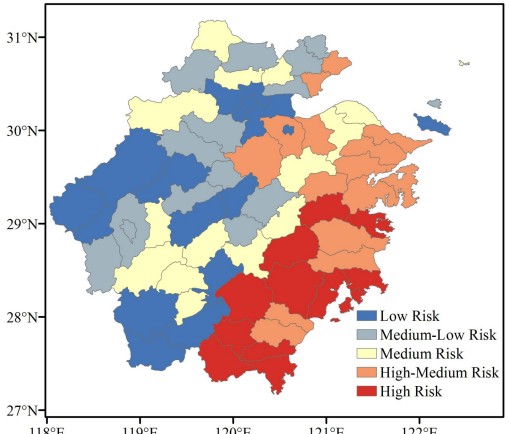


Fig. 11. Risk zoning of typhoon disasters in Zhejiang province.

## 5 Discussion and Conclusions

(1) An intensity index of factors causing typhoon disasters is developed, with highest values in
Wenzhou, Taizhou and Ningbo cities. A comparison between the distributions of the intensity index and
actual typhoon disasters in Zhejiang province from 2004 to 2012 shows that the index is a good
reflection of the possibility of typhoon disasters.
(2) Seven components are extracted after PCA of 29 variables affecting vulnerability. These seven
factors represent 81.9% of the total variance and are a good reflection of the index of population
vulnerability in Zhejiang province. Southwestern Zhejiang is the most vulnerable as it has a relatively
undeveloped economy, more mountainous areas and a higher risk of geological disasters.
Vulnerabilities are lower in cities as a result of better disaster prevention and reduction measures and a
better educated population.
(3) Typhoon disaster comprehensive risk index is obtained by combining the factors causing
typhoon disasters and population vulnerability. Based on the comprehensive risk index, risk zoning of
typhoon disasters in Zhejiang province is achieved. The southeast coastal areas are at high risk,
especially the boundary regions between Zhejiang and Fujian province, and Taizhou and Wenzhou

cities. The risk of typhoon disasters decreases quickly from coastal areas to inland regions. Cities are at medium to low risk because of their developed economy, high-quality houses and better educated population.

Although some interesting results have been obtained in this study, there are still some problems that require further study. As a result of the limited data on typhoon disasters, it is currently impossible to give a long time trend for high-resolution typhoon disaster analysis. It is also unclear whether this methodology can be applied to other regions. This paper mainly considers the effects of typhoon rain and typhoon wind, without considering the impact of storm surge. This is the limitation of the study, and we will explore the role of storm surges in future work.

## Acknowledgments

This study is supported by the Chinese Ministry of Science and Technology Project No. 2015CB452806.

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
