# Peer review of "Risk Zoning of Typhoon Disasters in Zhejiang Province, China"

_Natural Hazards and Earth System Sciences, 2018_

## Short Comment (SC1) · 28 Feb 2018

General Comments:

This paper analyses disasters made by tropical cycles and their reasons connecting with wind and rainfall. The study area is the Zhejiang province, east coastal province of China. For analyzing TC disaster, they develop an intensity index and population vulnerability index. They think the risk is defined to be included both intensity and population vulnerability index. Then they made zoning and regionalization for named comprehensive TC risk index. This is an interesting work, but the paper need to improve quality in order to meet the publishing standard of the NHESS.

Specific comments:

[Figure]

1, Please use word "Tropical Cycle" instead of Typhoon. 2, In your paper, you mentioned many times risk. In my knowledge, risk is future probability of hazard events. However, your paper studied the disaster events happened during 2004-2012. In this period, I would not like to mention risk. In your paper, you can only name disaster probability. 3, wind and precipitation data. Please clarify wind and precipitation data during TC events or not. 4, The figure should be re-redrawed as figures have no latitude and longitude. 5. Latest references should be cited.

---

## Referee Comment (RC1) · Anonymous Referee #1 · 25 Mar 2018

This study presents the risk mapping of Typhnoon disasters for a case study in Zhejiang, China. Such a study is of interest. However there are so many unknown issues in this manuscript, making it not readable. It is necessary to present , especially the mothology, in a much clearer manner. 1. In the "2. Study Area" section, it is better to give a background introduction of typhnoon disasters in the study area; otherwise, it is not unstoodable why you use Zhejiang Province as a case. 2. For the meteorogical data, how many stations are there in Zhejiang Province? The authors said 2419 stations provided by NMIC; it is not clear if they are distributed throughout China or only in Zhejiang province. It needs to be clarified. 3. Canonical Correlation Analysis is the main tool for this study, which is however not introduced to readers at all. It is necessary to give an introduction of this method and how it is applied in this study. 4. In

[Figure]

Section 3.2.4, it is necessary to introduce how the so-called SoVI is used to calculate the population vulnerability index. 5. The data source of typhoon disaster losses? 6. How is "typhoon rainstorm" defined? Its probability is an important issue in this study. How the probablity is determined? It is not introduced at all. 7. In Eq(1), it is not clear how population and economic loss are both included. Only for population or economic loss? Why in the form of "Ax+By", not a "multiplying" form? 8. P12L205, it says, after "performing factor analysis"..... How is the factor analysis performed? In the title of Table 3, principal component analysis is mentioned. How is the principal component analysis performed? Factor analysis is the principal component analysis? If these methods are used, they need to be introduced in the methodology section. 9. In Table 4, the derived disaster risk index are devided into 5 grades. How are the thoresholds are determined? It is not mentioned.

---

## Referee Comment (RC2) · Anonymous Referee #2 · 28 Mar 2018

As we known, researches on damage assessment and vulnerability is limited and difficult for the lack of precise data. This study discusses the vulnerability of the population and economic factors at the county level for Zhejiang Province, China. It is interesting. But there are still some questions and the manuscript is need to be improved. Questions are listed as follow: 1. The English of the manuscript is need be improved. 2. Extend data from 2012 to 2016? 3. The background of typhoon disaster over Zhejiang province must be introduced. 4. L74-75. Add reference about best track data from CMA. 5. Describe more detail about the OSAT methods. 6. L103: In this section, authors describe the methods of standardization. But corresponding variables are unknown. 7. Introduce the method of calculating probability of typhoon rainstorms. 8. L148: Do you sure about "over six sites"? 9. Durations of data in different parts are

different. Could durations of different kinds of data are uniformed? 10. L178: What criterions are used in calculating precipitation or wind? 11. L181: The distribution of sample sizes? 12. L194: the intensity index is calculated for each typhoons and each stations? And how get the distribution of intensity indexes in figure 8? 13. L199-L238. I am confused by the section 5.2. Each factor or component in factor 1 to factor7 in equation 2 or seven components in table 3 is a vector with 29 variables? Please describe more details about factors or components. The signs of vectors from PCA maybe opposite of real meaning. Do you do detail analysis on the relations of variables in a vector and between vectors? The equation 2 is must be evaluated carefully. 14. Could you find data of 29 variables in table 2 in early year? If the difference of population vulnerability between 2010 to early year is analyzed, the manuscript will to be more valuable.

---

## Author Comment (AC1) · 20 Jun 2018

Dear Editor and the reviewer,

We do appreciate your constructive, thoughtful, careful, and helpful comments and suggestions. After careful discussions, calculations, and analyses, we finished the preparation of responses to you. The responses are structured in 3 sequence: (1) comments from Referees, (2) author's response, (3) author's changes in manuscript. If there are any new comments or suggestions, please let us know.

Best Regards

Yi Lu and the coauthors

Response to Short Comments

1. Please use word "Tropical Cycle" instead of Typhoon.

Reply: Thanks for your comment. Considering to be consistent with some published researches (Chen, 2007; Chen et al., 2011; Ding et al., 2002; Niu et al., 2011), we tend to introduce a definition of typhoon rather than change the name. At the beginning of "Introduction", "Tropical cyclones cause" has been changed into "Typhoon, which means tropical cyclone in this paper, often causes".

2. In your paper, you mentioned many times risk. In my knowledge, risk is future probability of hazard events. However, your paper studied the disaster events happened during 2004-2012. In this period, I would not like to mention risk. In your paper, you can only name disaster probability.

Reply: We do appreciate the comment. We agree and understand that risk is future probability of hazard events. However, suppose future probability is the same as historical probability for a specific period, we can understand risk by learning from past events. We tend to make a specific statement about this rather to change "risk" into "disaster probability".

According to the comments, modifications include:

(1)A sentence "As risk is future probability of hazard events, when suppose future probability is the same as historical probability for a specific period, we can understand risk by learning from past events." is added and becomes the first sentence of "Abstract".

(2)A sentence "For risk analyses of typhoon precipitation and typhoon wind (please see detail in sections 3.1 and 3.2), suppose future probability is the same as historical probability, we then select the period of 1980 – 2014." is added as the first sentence of the third paragraph in section 2.1.1.

3. Wind and precipitation data. Please clarify wind and precipitation data during TC events or not.

Reply: Thanks for your suggestion. As described in 3.1.1, the original daily precipitation data and maximum wind speed in this paper are from the National Meteorological Information Center. Then we distinguish typhoon precipitation and winds by using the OSAT method (3.2.1). Therefore, all typhoon wind and rain mentioned later in this paper were during TC events.

To clarity wind and precipitation data, we have added following sentences at the end of section 3.2.1. "With the application of the OSAT method, daily precipitation and wind data over the mainland of China during 1980 to 2014 are used for identifying typhoon precipitation and wind data.".

4. The figure should be redrawed as figures have no latitude and longitude.

Reply: Thanks for your suggestion. Figures have been modified according to the suggestion.

5. Latest references should be cited.

Reply: Thanks for your suggestion. According to the suggestion, we have added some latest references in "1 Introduction " and "Reference". At the end of line 44, we add following sentences. "Recently, some research built quantitative assessment in some provinces and carried out preliminary studies on pre-evaluating typhoon disasters (Huang and Wang, 2015; Yin and Li, 2017)." At the end of line 56, we add following sentences. "Xu et al. (2015) comprehensively assessed the impact of typhoons across China using the geographical information system. The future direction of tropical cyclone risk management is quantitative risk models (Chen et al., 2017)."

Added references:

Chen, W. F., Duan, Y. H., and Lu, Y.: Review on Tropical Cyclone Risk Assessment, Journal of Catastrophology, 32(4), 2017. (in Chinese)

Huang, W. K. and Wang, J. J.: Typhoon damage assessment model and analysis in Taiwan,ăNatural Hazards,ă79(1), 497-510, 2015.

Xu, X., Sun, D., and Guo, T.: A systemic analysis of typhoon risk across china, Natural Hazards, 77(1), 461-477, 2015.

Yin, Y. Z. and Li, H. L.: Preliminary study on pre-evaluation method of typhoon disaster in China, Meteorological Monthly, 43(6):716-723, 2017. (in Chinese)

Please also note the supplement to this comment:
https://www.nat-hazards-earth-syst-sci-discuss.net/nhess-2018-14/nhess-2018-14-AC1-supplement.pdf

---

## Author Response (AR1)

Dear Editor and the reviewers,

We do appreciate your constructive, thoughtful, careful, and helpful comments and suggestions.

After careful discussions, calculations, and analyses, we finished the preparation of responses to you. There are totally four parts: "Response to Reviewer 1", "Response to Reviewer 2",

"Response to Short Comments" and "Risk Zoning of Typhoon Disasters in Zhejiang Province,

China" with tracing.

If there are any new comments or suggestions, please let us know.

Best Regards

Yi Lu and the coauthors

**Response to Reviewer 1**

1. In the "2. Study Area" section, it is better to give a background introduction of typhoon disasters in the study area; otherwise, it is not unstoodable why you use Zhejiang Province as a case.

Reply: We do appreciate the helpful suggestion.

Mainly according to the suggestion, some modifications have been done. The modifications include:

(1) The structure of the paper has been changed. Considering that Section 2 is so thin and unbalanced to other sections, "2 Study Area" is merged into "3 Data and Methods" and then the

Section numbers hereafter are changed.

(2) In the last paragraph of "Introduction", a sentence "In this study, Zhejiang province, which is frequently affected by the strongest landfall typhoons (Ren et al., 2008) and experiences most serious typhoon disasters (Liu and Gu, 2002) in the mainland of China, is selected as the study area." has been added.

The following two new references have been added:

Liu, T. J. and Gu, J. Q.: A statistical analysis of typhoon disasters in Zhejiang province, Journal of

Catastrophology, 17 (4): 64-71, 2002. (in Chinese)

Ren, F. M., Wang, X. L., Chen, L. S., and Wang, Y. m.: Tropical cyclones landfalling on mainland

China, Hainan and Taiwan and their correlations, Acta Meteorologica Sinica, 66 (2): 224-235,

2008. (in Chinese)

2. For the meteorogical data, how many stations are there in Zhejiang Province? The authors said

2419 stations provided by NMIC; it is not clear if they are distributed throughout China or only in

Zhejiang province. It needs to be clarified.

Reply: Thanks so much for the comment. In this paper, the OSAT method need to identify typhoon wind and precipitation from wide range than Zhejiang province, so 2419 stations of precipitation data and 2479 stations of wind data over the mainland of China are used, which all contained 71

stations corresponding to counties in Zhejiang province.

We add following sentences at the end of L80 in original section 3.1.1.

"In addition, the OSAT method need to identify typhoon wind and precipitation from a wider range than Zhejiang province (please see detail in section 2.2.1), so 2419 stations of precipitation data and 2479 stations of wind data over the mainland of China are used, which all contained 71

stations corresponding to counties in Zhejiang province."

3. Canonical Correlation Analysis is the main tool for this study, which is however not introduced to readers at all. It is necessary to give an introduction of this method and how it is applied in this study.

Reply: Thanks for your suggestion. According to the suggestion, we have added introduction and application of Canonical Correlation Analysis (CCA) in this paper.

We add following sentences at the end of L102 in original section 3.2.2.

"In statistics, canonical correlation analysis (CCA) is a way of inferring information from cross-covariance matrices. If we have two vectors X = (X1, ..., Xn) and Y = (Y1, ..., Ym) of random variables, and there are correlations among the variables, then CCA can find linear combinations of the Xi and Yj which have maximum correlation with each other (Hardoon et al., 2014). The method was first introduced by Hotelling in 1936 (Hotelling, 1936). The main point of CCA is to separate linear combination of new variables from the two sets of variables. In this case, the correlation coefficient between new variables reaches the maximum. In this paper, we chose factors causing typhoon disasters as a set of variables, and typhoon disaster as another. Under the maximum canonical correlation coefficient, the linear combination coefficients (typical variable coefficients) of factors causing typhoon disasters can be used as weight coefficients of this group of variables. Then we can determine the impact of factors causing typhoon disasters."

The added references are as follows:

Hardoon, D. R., Szedmak, S., and Shawetaylor, J.: Canonical Correlation Analysis: An Overview with Application to Learning Methods, Neural Computation, 16(12):2639-2664, 2014.

Hotelling, H.: Relations between two sets of variates, Biometrika, 28(3/4), 321-377, 1936.

4. In Section 3.2.4, it is necessary to introduce how the so-called SoVI is used to calculate the population vulnerability index.

Reply: Thanks very much for the suggestion. According to the suggestion, we have added introduction and application of SoVI in this paper.

We add following sentences at the beginning of L109 in original section 3.2.4.

"County-level socioeconomic and demographic data are used to construct an index of social vulnerability to environmental hazards named the Social Vulnerability Index (SoVI). Principal Component Analysis (PCA) is the primary statistical technique for constructing the SoVI. The PCA method captures multi-dimensionality by transforming the raw dataset to a new set of independent variables. Then a few components can represent the dimensional data, and underlying factors can be identified easily. These new factors are placed in an additive model to compute a summary score—SoVI (Cutter et al., 2003)."

The added reference is as follows:

Cutter, S. L., Boruff, B. J., and Shirley, W. L.: Social Vulnerability to Environmental Hazards, Social Science Quarterly, 84(2):242-261, 2003.

5. The data source of typhoon disaster losses?

Reply: Typhoon disaster losses used in this paper are obtained from the National Climate Center who collected these disaster data from local meteorological departments. The data source can be seen in lines 82-83 of original section 3.1.2.

6. How is "typhoon rainstorm" defined? Its probability is an important issue in this study. How the probability is determined? It is not introduced at all.

Reply: Thanks very much for the comment and suggestion. According to the suggestion, we have added a definition of "typhoon rainstorm" and "typhoon torrential rainstorm". In original section 4.1, we did some research about risk of typhoon rainstorm. Typhoon rainstorm in this study means daily typhoon precipitation over 50mm, and typhoon torrential rainstorm means daily typhoon precipitation over 100mm. The probability is the annual possibility of the occurrence of typhoon rainstorms. The probability denominator is the total number of years, and the numerator is the annual frequency of typhoon precipitation. If a station experiences typhoon precipitation in one year, the numerator increases by one.

We add following sentences at the end of original L133 before "Based on".

"Typhoon rainstorm in this study means daily typhoon precipitation over 50mm, and typhoon torrential rainstorm means daily typhoon precipitation over 100mm. The probability is the annual possibility of the occurrence of typhoon rainstorms."

7. In Eq(1), it is not clear how population and economic loss are both included. Only for population or economic loss? Why in the form of "Ax+By", not a "multiplying" form?

Reply: Thanks for the comment. As we explained in question 3, the main point of CCA is to separate linear combination of new variables from the two sets of variables. In this paper, we chose typhoon disaster-causing factors as a set of variables, and typhoon disaster as another. Typhoon disaster-causing factors (typhoon wind and precipitation) and typhoon disasters (affected population and economic loss) are both contained. When the typical correlation coefficient pass the significance test, weight discrimination can be made to determine A and B. Both typhoon rainstorm and high wind will bring certain disasters. When reaching a certain critical value, they will have a superposition effect. However, the effects of wind and rain on disaster are different. Therefore, it is necessary to determine their influence through typical correlation analysis, which is a typical variable coefficient. So the form is "Ax+By", not a multiplier.

8. P12 L205, it says, after "performing factor analysis"..... How is the factor analysis performed? In the title of Table 3, principal component analysis is mentioned. How is the principal component analysis performed? Factor analysis is the principal component analysis? If these methods are used, they need to be introduced in the methodology section.

Reply: Thanks very much for the comment and suggestion. The factor analysis performed here is Principal Component Analysis (PCA), which is the primary statistical procedure for constructing the SoVI. According to the suggestion, we have added detailed explanations of PCA in original section 3.2.4, which have been answered in question 4.

We add following sentences at the beginning of L109 in original section 3.2.4.

"County-level socioeconomic and demographic data are used to construct an index of social vulnerability to environmental hazards named the Social Vulnerability Index (SoVI). Principal Component Analysis (PCA) is the primary statistical technique for constructing the SoVI. The PCA method captures multi-dimensionality by transforming the raw dataset to a new set of independent variables. Then a few components can represent the dimensional data, and underlying factors can be identified easily. These new factors are placed in an additive model to compute a summary score—SoVI (Cutter et al., 2003). "

9. In Table 4, the derived disaster risk index are divided into 5 grades. How are the thoresholds are
determined? It is not mentioned.
Reply: Thanks for the comment. The classification of typhoon disaster risk index is based on the
natural breaks method (Jenks) provided by Arcgis. Then we divide disaster risk index into 5
grades, which represent five risk zones for typhoon disasters in Zhejiang province.
We add following sentences at the end of original L248.
"The classification of typhoon disaster risk index is based on the natural breaks method (Jenks)
provided by Arcgis."

# Response to Reviewer 2

1. The English of the manuscript is need be improved.
Thanks for your comment. According to the suggestion, modifications include:
(1) The first half of Abstract has been rewritten.
(2) "Province" has been changed into "province".
(3) "Comprehensive Risk Index for Typhoon Disasters" has been changed into "Typhoon Disaster
Comprehensive Risk Index".
(4) The second half of the last paragraph in Introduction has been rewritten.
(5) The structure of the paper has been changed. Considering that Section 2 is so thin and
unbalanced to other sections, "2 Study Area" is merged into "3 Data and Methods" and then
the Section numbers hereafter are changed.
(6) Analyses for the figures have been revised especially these for Figure 7.
(7) For grammar, all past tense have been changed into present tense.
(8) Other specific modifications can be seen in detail in the text with revision-tracing.

2. Extend data from 2012 to 2016?
Thanks very much for the suggestion.
As county-level typhoon disaster data is so limited and it's hard to get new data, we can't extend
data from 2012 to 2016.

3. The background of typhoon disaster over Zhejiang province must be introduced.
According to the suggestion, modifications include:
(1) In the last paragraph of "Introduction", a sentence "In this study, Zhejiang province, which is
frequently affected by the strongest landfall typhoons (Ren et al., 2008) and experiences most
serious typhoon disasters (Liu and Gu, 2002) in the mainland of China, is selected as the study
area." has been added.
(2) The following two new references have been added:
Liu, T. J. and Gu, J. Q.: A statistical analysis of typhoon disasters in Zhejiang province, Journal of
Catastrophology, 17 (4): 64-71, 2002. (in Chinese)
Ren, F. M., Wang, X. L., Chen, L. S., and Wang, Y. m.: Tropical cyclones landfalling on mainland
China, Hainan and Taiwan and their correlations, Acta Meteorologica Sinica, 66 (2): 224-235,
2008. (in Chinese)

4. L74-75. Add reference about best track data from CMA.

Reply: Thanks for your suggestion. According to the suggestion, we have added references about best track data from CMA, and cite them in original section 3.1.1.

New references are added in "References".

Eunjeong, C. and Ying, M.: Comparison of three western North Pacific tropical cyclone best track datasets in seasonal context, Journal of the Meteorological Society of Japan, 89(3):211-224, 2009.

Li, S. H. and Hong, H. P.: Use of historical best track data to estimate typhoon wind hazard at selected sites in China, Natural Hazards, 76(2):1395-1414, 2015.

5. Describe more detail about the OSAT methods.

Reply: Thanks very much for the suggestion. According to the suggestion, we have added more detail about the OSAT method.

We add following sentences in L96 of original section 3.2.1 before " Lu".

"The OSAT method is a numerical technique to separate tropical cyclone induced precipitation from adjacent precipitation areas. Based on the structural analysis of precipitation field, it can be divided into different rain belts. Then, according to the distances between a TC center and these rain belts, typhoon center and each station, typhoon precipitation is distinguished."

6. L103: In this section, authors describe the methods of standardization. But corresponding variables are unknown.

Reply: Thanks for the comment. According to the comment, we have added introduction of corresponding variables.

We add following sentences at the end of L107 in original section 3.2.3.

"Z-score standardization is used for calculating intensity index of factors causing typhoon disasters. Both typhoon precipitation and typhoon maximum wind speed are standardized by this method. When calculating typhoon disaster comprehensive risk index (R), we use MIN-MAX

standardization to standardize the intensity index of the factors causing typhoon disasters (I) and the population vulnerability index (SoVI)."

7. Introduce the method of calculating probability of typhoon rainstorms.

Reply: Thanks very much for the suggestion. According to the suggestion, we have added a definition of "typhoon rainstorm" and "typhoon torrential rainstorm". In original section 4.1, we did some research about risk of typhoon rainstorm. Typhoon rainstorm in this study means daily typhoon precipitation over 50mm, and typhoon torrential rainstorm means daily typhoon precipitation over 100mm. The probability is the annual possibility of the occurrence of typhoon rainstorms. The probability denominator is the total number of years, and the numerator is the annual frequency of typhoon precipitation. When a station has experienced typhoon precipitation in one year, the numerator increases by one.

We add following sentences at the end of original L133 before "Based on".

"Typhoon rainstorm in this study means daily typhoon precipitation over 50mm, and typhoon torrential rainstorm means daily typhoon precipitation over 100mm. The probability is the annual possibility of the occurrence of typhoon rainstorms. "

8. L148: Do you sure about "over six sites"?

Reply: Thanks for your question. We feel sorry for that it is a translation error. This should be referred to Typhoon wind over 6 grade ($\geq$10.8 m/s). Figure 6 shows average duration (days) of typhoon winds at each station in Zhejiang province from 1980 to 2014. We have modified original L148.

Original L148 is modified as follows.

"The average duration (days) of typhoon winds (over 6 grade) is calculated in Zhejiang province (Figure 6). "

9. Durations of data in different parts are different. Could durations of different kinds of data are uniformed?

Reply: Thanks for the comment. In the original paper, we used three durations of data, including typhoon precipitation during 1960 - 2013, typhoon wind during 1980 - 2014, and typhoon disaster data during 2004 - 2012. According to the suggestion, durations of daily precipitation and wind have been uniformed with 1980–2014. Duration of typhoon disaster data remains unchanged. Detailed reasons are as follows.

(1) First of all, because of limited access to county-level typhoon disaster data, we have only obtained data during 2004 to 2012 from National Climate Center. So all analyses of intensity index of factors causing typhoon disasters are during 2004 to 2012, which remains unchanged. However, this duration is short for risk analyses of typhoon precipitation and typhoon wind. Therefore, longer time-series data are needed.

(2) We feel sorry for that it is a expression mistake to say "The statistics showed a rapid increase in the number of automated wind measurement stations from 1980" in original L77. As Lu et al. (2016) mentioned, considering the homogeneity and continuity of wind data, we use daily wind data during 1980 - 2014 to to identify typhoon wind.

(3) Considering the consistency between wind and precipitation data, 1980 to 2014 is selected as the period of study. In addition, the OSAT method need to identify typhoon wind and precipitation from wide range rain belts, so 2419 stations of precipitation data and 2479 stations of wind data over the mainland of China are used, which all contained 71 stations corresponding to counties in Zhejiang province.

According to the suggestion, modifications include:

(1) The introduction of daily precipitation and wind data in original section "3.1.1 Typhoon, Precipitation and Wind Data" are rewrote as follow.

"Daily precipitation data for 2479 stations and daily wind data for 2419 stations during the time period 1960 - 2014 over the mainland of China are obtained from National Meteorological Information Center. The maximum wind speed is given as the maximum of 10-minute mean. In this paper, two time periods of precipitation and wind data are used.

Because of limited access to county-level typhoon disaster data, we have only obtained data during 2004 to 2012. So when calculating intensity index of factors causing typhoon disasters, time period of typhoon precipitation and typhoon wind are the same as typhoon disasters, which is 2004 - 2012.

For risk analyses of typhoon precipitation and typhoon wind (please see detail in sections 3.1 and 3.2), suppose future probability is the same as historical probability, we then select the period of 1980 – 2014. As Lu et al. (2016) mentioned, considering the homogeneity of wind data, we use the period of 1980 - 2014 for wind analysis. To ensure the consistency between wind and precipitation data, 1980 - 2014 is selected as the period. In addition, the OSAT method need to identify typhoon wind and precipitation from a wider range than Zhejiang province (please see detail in section 2.2.1), so 2419 stations of precipitation data and 2479 stations of wind data over the mainland of China are used, which all contained 71 stations corresponding to counties in Zhejiang province."

(2) The analyses in original section "4.1 Risk of Typhoon Rainstorms" are modified. All time periods in this section have been changed to 1980 - 2014, with corresponding changes of calculations and pictures.

10. L178: What criterions are used in calculating precipitation or wind?

Reply: Thanks for the comment. If a typhoon disaster occurs and there is a corresponding typhoon wind or typhoon precipitation, it will be included in the sample.

11. L181: The distribution of sample sizes?

Reply: Good comment. The total valid disaster records of Zhejiang province from 2004 to 2012 are 421. To establish an intensity index of typhoon disaster-causing factors, we carry out CCA analysis. Taking the county as a unit, we select all the typhoons that affected the population, which means all records with an affected population greater than 0. The total precipitation and daily maximum wind speed during affected typhoons measured in each county are used. The total sample size is 322. Then, we do CCA analyses for all the typhoons that caused direct economic losses in the same way, and the total sample size is 404.

According to the suggestion, modifications include:

(1) We add following sentences at original L179 after "factors causing typhoon disasters".

" The total sample size is 322. ".

(2) We add following sentences at original L182 before "(Table 1)".

", and the total sample size is 404.".

12. L194: the intensity index is calculated for each typhoons and each stations? And how get the distribution of intensity indices in figure 8?

Reply: Thanks for the comment. The intensity index is calculated for each typhoon at each station. Then we average all intensity indices at each station, and we can get the distribution of intensity indices in figure 8.

13. L199-L238. I am confused by the section 5.2. Each factor or component in factor 1 to factor7 in equation 2 or seven components in table 3 is a vector with 29 variables? Please describe more details about factors or components. The signs of vectors from PCA maybe opposite of real meaning. Do you do detail analysis on the relations of variables in a vector and between vectors? The equation 2 is must be evaluated carefully.

Reply: Thanks very much for the comments and suggestions.

(1) For the first question, component 1 to component 7 are vectors in table 3 with 29 variables. After performing PCA of the 29 variables, 7 components with eigenvalue equal to or greater than 1 are extracted.

(2) For the second question, 7 components are examined manually as to whether they increase (+)

or decrease (−) vulnerability and they are assigned a cardinality on that basis. Then the
vulnerability index is produced by summing all the components using equal weighting, following
the Chen (2013) approach.
To conveniently describe details about 7 components, we rename 29 variables (Table 2). After
PCA, we obtain 7 components. The signs and contained variables of 7 components are shown in
Table 3. We can see 6 components increase (+) vulnerability and a component decrease (−)
vulnerability. For example, the first component, which reflects the income of the population and
the employment situation, is positive because the more property there is in an area, the higher the
vulnerability to damage. The second component, which reflects education level of the population,
is negative because if education level is higher, then the population's awareness of disaster
prevention and reduction is greater and their vulnerability is lower.
According to the comments and suggestions, modifications include:
(1) We rename the 29 variables in Table 2.
Table 2. The 29 variables affecting vulnerability in Zhejiang province.

| | variables | Name |
|---|---|---|
| 1 | Per capita disposable income of urban residents (yuan) | UBINCM |
| 2 | Percentage of female (%) | QFEMALE |
| 3 | Percentage of minority (%) | QMINOR |
| 4 | Median age | MEDAGE |
| 5 | Unemployment rate (calculated - unemployed population / (unemployed + total population) | QUNEMP |
| 6 | Population density | POPDEN |
| 7 | Percentage of urban population (%) | QUBRESD |
| 8 | Percentage of non-agricultural household population (%) | QNONAGRI |
| 9 | Percentage of households that living in rented houses (%) | QRENT |
| 10 | Percentage of employees working in primary industries and mining (%) | QAGREMP |
| 11 | Percentage of employees working in secondary industries (%) | QMANFEMP |
| 12 | Percentage of employees working in tertiary industries (%) | QSEVEMP |
| 13 | Household size (person / household) | PPUNIT |
| 14 | Percentage of population with college degree (25 years old and older) | QCOLLEGE |
| 15 | Percentage of population with high school degree (20 years old and older) | QHISCH |
| 16 | Percentage of illiterate people (15 years old and older) | QILLIT |
| 17 | Population growth rate (2000-2010) | POPCH |
| 18 | Average number of rooms per household (inter / household) | PHROOM |

| 19 | Per capita housing construction area (m² / person) | PPHAREA |
| 20 | Percentage of premises without tap water (%) | QNOPIPWT |
| 21 | Percentage of premises without a kitchen (%) | QNOKITCH |
| 22 | Percentage of premises without a toilet (%) | QNOTOILET |
| 23 | Percentage of premises without a bath (%) | QNOBATH |
| 24 | Number of beds per 1000 person in health care institutions | HPBED |
| 25 | Number of medical personnel per 1000 resident population | MEDPROF |
| 26 | Percentage of people under 5 | QPOPUD5 |
| 27 | Percentage of population over 65 years old | QPOPAB65 |
| 28 | Population dependency ratio (%) | QDEPEND |
| 29 | Percentage of population covered by subsistence allowances (%) | QSUBSIST |

(2) We add contained variables of 7 components with different signs in Table 3.

               Table 3. The seven components extracted by PCA.

| Components | Contained variables | Name | (Sign) |
|---|---|---|---|
| 1 | QMANFEMP, UBINCM, QAGREMP, QRENT, POPCH, QDEPEND, QSUBSIST, QPOPAB65, POPDEN, MEDAGE, QNOKITCH, QILLIT, PHROOM, PPHAREA | Employment and poverty | (+) |
| 2 | QHISCH, QCOLLEGE, QNONAGRI, QSEVEMP, HPBED, MEDTECH | Education | (-) |
| 3 | QNOBATH, QNOTOILET, PPUNIT | Number of dilapidated houses | (+) |
| 4 | QILLIT, QDEPEND, QPOPUD5, MEDAGE | Illiteracy and juvenile population | (+) |
| 5 | QFEMALE, PHROOM, PPHAREA, QSEVEMP | Household size and ratio of women | (+) |
| 6 | QMINOR | Ethnic minority | (+) |
| 7 | QUNEMP, QNOPIPWT | Unemployment and housing size | (+) |

(3) To distinguish the 7 components and 29 variables more clearly, we replace all "factor" with "component" in this section.

14. Could you find data of 29 variables in table 2 in early year? If the difference of population vulnerability between 2010 to early year is analyzed, the manuscript will to be more valuable.

Reply: Thanks for your thoughtful suggestion. The population data used in this paper is obtained from the sixth national population census of the Population Census Office of the National Bureau of Statistics of China and the 2010 statistical yearbooks of each city in Zhejiang province published by the cities' statistical bureaus. There exist many missing and abnormal records in the original data, which take a long time to be processed. This article focuses on typhoon disaster risk zoning in Zhejiang province, so we didn't discuss the difference of population vulnerability between 2010 to early year. The variation of population vulnerability is an interesting topic. Maybe we can discuss it in future work.

**Response to Short Comments**

1. Please use word "Tropical Cycle" instead of Typhoon.

Reply: Thanks for your comment. Considering to be consistent with some published researches (Chen, 2007; Chen et al., 2011; Ding et al., 2002; Niu et al., 2011), we tend to introduce a definition of typhoon rather than change the name. At the beginning of "Introduction", "Tropical cyclones cause" has been changed into "Typhoon, which means tropical cyclone in this paper, often causes".

2. In your paper, you mentioned many times risk. In my knowledge, risk is future probability of hazard events. However, your paper studied the disaster events happened during 2004-2012. In this period, I would not like to mention risk. In your paper, you can only name disaster probability.

Reply: We do appreciate the comment.

We agree and understand that risk is future probability of hazard events. However, suppose future probability is the same as historical probability for a specific period, we can understand risk by learning from past events. We tend to make a specific statement about this rather to change "risk" into "disaster probability". According to the comments, modifications include:

(1) A sentence "As risk is future probability of hazard events, when suppose future probability is the same as historical probability for a specific period, we can understand risk by learning from past events." is added and becomes the first sentence of "Abstract".

(2) A sentence "For risk analyses of typhoon precipitation and typhoon wind (please see detail in sections 3.1 and 3.2), suppose future probability is the same as historical probability, we then select the period of 1980 – 2014." is added as the first sentence of the third paragraph in original section 3.1.1.

3. Wind and precipitation data. Please clarify wind and precipitation data during TC events or not.

Reply: Thanks for your suggestion. As described in original section 3.1.1, the original daily precipitation data and maximum wind speed in this paper are from the National Meteorological Information Center. Then we distinguish typhoon precipitation and winds by using the OSAT

method (original section 3.2.1). Therefore, all typhoon wind and rain mentioned later in this paper were during TC events.

To clarity wind and precipitation data, last sentence in original section 3.2.1 is rewrote as follows.

"With the application of the OSAT method, daily precipitation and wind data over the mainland of China during 1980 to 2014 are used for identifying typhoon precipitation and wind data.".

4. The figure should be redrawed as figures have no latitude and longitude.

Reply: Thanks for your suggestion. Figures have been modified according to the suggestion.

5. Latest references should be cited.

Reply: Thanks for your suggestion. According to the suggestion, we have added some latest references in "1 Introduction " and "References".

At the end of original line 44, we add following sentences.

[revised manuscript text omitted]

This study wasis carried out in Zhejiang Provinceprovince (Figure 1) and includinged 11 cities along the Yangtze River Delta. Zhejiang Provinceprovince is in the eastern part of the East China Sea and south ofto Fujian Provinceprovince, which is one of the most economically powerful provinces in

China.

[Figure]

[Figure]

Figure 1. Maps of Zhejiang province, China showing location and major cities.

** Data and Methods**

**2.1 Data**

**2.1.1 Typhoon, Precipitation and Wind Data**

The typhoon data used in this study are the best-track tropical cyclone datasets from  Shanghai

Typhoon Institute for the time period 1960 - 2014 (Eunjeong and Ying, 2009; Li and Hong, 2015).

Daily precipitation data for 2479 stations and daily wind data for 2419 stations during the time period

1960 - 2014 over the mainland of China are obtained from  National Meteorological

Information Center. The maximum wind speed is given as the maximum of 10-minute mean. In this paper, two time periods of precipitation and wind data are used.

Because of limited access to county-level typhoon disaster data, we have only obtained data during 2004 to 2012. So when calculating intensity index of factors causing typhoon disasters, time period of typhoon precipitation and typhoon wind are the same as typhoon disasters, which is 2004 -

2012.

For risk analyses of typhoon precipitation and typhoon wind (please see detail in sections 3.1 and

3.2), suppose future probability is the same as historical probability, we then select the period of 1980 –

2014. As Lu et al. (2016) mentioned, considering the homogeneity of wind data, we use the period of

1980 - 2014 for wind analysis. To ensure the consistency between wind and precipitation data, 1980 -

2014 is selected as the period. In addition, the OSAT method need to identify typhoon wind and precipitation from a wider range than Zhejiang province (please see detail in section 2.2.1), so 2419

stations of precipitation data and 2479 stations of wind data over the mainland of China are used, which all contained 71 stations corresponding to counties in Zhejiang province.

**3̶2.1.2 Disaster and Social Data**

Disaster data for each typhoon that affected Zhejiang P̶r̶o̶v̶i̶n̶c̶e̶province from 2004 to 2012 w̶e̶r̶e̶are obtained from the National Climate Center and the number of records for each county is shown in

Figure 2. Of the 11 cities in Zhejiang P̶r̶o̶v̶i̶n̶c̶e̶province, Wenzhou and Taizhou record̶e̶d̶ the most typhoon disasters, with a maximum beingo̶f̶ 17. Fewer typhoon disasters w̶e̶r̶e̶
[revised manuscript text omitted]

---

## Referee Report (RR1)

**Journal NHESS Referee Report for "Risk Zoning of Typhoon Disasters in Zhejiang Province, China" by Yi Lu et al.**

**General Comments:**

1. Manuscript required a lot of editing for proper English language usage. Some of this reviewer's recommended changes may have changed the intent of the authors. If that is the case, then the sentence or paragraph should be rewritten.

2. The study mentions storm surge as one of the three factors affecting "typhoon" disasters, yet there is no assessment of the effects of storm surge. Storm surge kills many more people than wind. If wind intensity is used as a proxy for storm surge, that should be indicated. Tropical cyclone size is an important consideration for storm surge and is sometimes more important than intensity. This should be mentioned as a limitation of the study.

3. The first line of the Introduction equates "typhoon" to "tropical cyclone". The term "tropical cyclone" generally includes tropical depressions, tropical storms, and typhoons. This should be clarified at the beginning.

4. How do you define a disaster? What are the criteria?

5. For consistency, use either "indices" or "indexes".

**Specific Comments:**

**1. Abstract:**

a. Lines 463-464: Doesn't make sense: Do you mean "Assuming that risk signifies probability of hazard events and that future probability is the same as historical probability for a specific period, we can understand risk by learning from past events." If not, p[lease rewrite the sentence.

b. Line 465: "…over mainland China during 1980-2014 and disaster and …"

c. Line 470: "The above analyses…"

d. Line 474: Locate Hangzhou Bay on your map.

e. Lines 475-477: Move this sentence to the end of the paragraph (after line 482).

**2. Introduction:**

a. Lines 486-487: See Iten 3 under General Comments.

b. Line 490: Delete "the".

c. Line 497: Delete "had".

d. Line 504:  I believe that this reference is for R. A. Pielke, Jr. (not his father Sr.). Check this out here and at lines 857-860 in the reference section.

e.  Lines 521: Consider making two separate sentences: "…disaster  loses.  Few studies…"

**3.  Data and Methods:**

a.  Line 531:  Do you mean "north of" instead of "south of"?

b.  Line 534 and Figure 1:  Identify Hangzhou Bay on the map and add to Legend.  The legend is not as clear as stating "(in red)" after 'major cities" and add "in provinces (in black)".  Also, does the scale need to be in kilometers as well as in miles?

c.  Line 540:  "…from the National…"

d.  Line 544:  ";;;from 2004 to 2012. So when calculating the intensity index of factors causing typhoon disasters, the time…"

e.  Lines 547-549:  Do you mean:  "…(see details in section 3.1 and 3.2), we assume that future probability is the same…"?  Otherwise, please rewrite.

f.  Line 551:  "…period for both.  In addition, the Objective Synoptic Analysis Technique (OSAT) method…"

g.  Line 552:  "…details…"

h.  Line 559:  "…being 17 at XXXX" ; add either Wenzhou or Taizhou.  Also, se General comments Item 4.

i.  Where is "Quzhou";  indicate on the map.

j.  Figure 2, line 566:  See comment in General Comments Line 4.

k. Line 572:  "…structural analysis of a precipitation…

l.  Line 584: "…combinations…"

m. Line 595:  "…calculating the intensity…"

n.  Line 597:  "…calculating the typhoon…"

o.  Lines 592-599:  Reference "MIN-MAX standardization" and "Z-score standardization" techniques' Explain rationale for selecting the Z-score technique as you did for the MAX-MIN technique.

p.  Lines 607-608:  Consider:  "Based on various SoVIs derived for disaster social vulnerability in America, Chen et al. (2014) selected 29 variables as proxies…"

q.  Line 609:  Consider:  "….  We then use these vulnerability indexes to calculate the population…"

**4. Typhoon Disaster Losses and Factors:**

a.  Page 19:  Discuss Figure 3c and 3d in the text.  Do not need the term "unit" in the caption.

b.  Line 630:  Locate "Quzhou city" on the map.

c.  Line 632:  "…typhoon disasters, "typhoon rainstorm" means…and "torrential rainstorm" means…"

d.  Lines 652-661:  I assume that "grades" pertains to some kind of typhoon wind-damage scale.  Is that correct.  In any event, you need to define wind values for Grades 6, 8, 10, and 12.

e.  Line 661:  "…and are only…"

f.  Line 663:  Do you mean:  "…with those with a high risk of typhoon…"

g.  Line 667:  Rewrite:  "…typhoon winds of (a) grade 6 or above, (b) grade 8 or above, (c) grade 10 or above, and (d) grade 12 or above."

**5.  Risk Assessment and Regionalization of Typhoon Disasters**

a.  Line 676:  Are factors and hazards used interchangeably? Storm surge is mentioned again as a "main factor", yet there is not storm surge losses or assessments in then study.

b.  Line 680:  "…includes…"

c.  Table 1, line 690, first row of Table:  Spelling/typo:  "Disasters"

d.  Lines 699-700:  "…and Ningbo cities are identified, which is consistent with the nresults of Chen et al. (2011).…"

e.  Line 708, Table 2:  Define how "primary", "secondary' and "tertiary" industries differ.

f.  Lines 744 and 766, Figures 10 and 11:  Why is the island northeast of Ningbo city of "low risk", while Ningbo is medium to high risk?

g.  Lines 777-778:  "…and a better educated population."

h.  Line 787:  "…further study…."

**6  References:**

a.  Lines 798, 846, and 888: Capitalize:  "China".

b.  Line 859:  Capitalize "United States"

c. Lines 857-860  Check Pielke Jr vice Pielke (the father).

---

## Author Response (AR4)

Dear Editor and the reviewers,

We do appreciate your constructive, thoughtful, careful, and helpful comments and suggestions. After careful discussions, calculations, and analyses, we finished the preparation of responses to you. There are totally three parts: "Response to Reviewer 1", "Response to Reviewer 3", and a marked-up version of the manuscript "Risk Zoning of Typhoon Disasters in Zhejiang Province, China".

If there are any new comments or suggestions, please let us know.

Best Regards

Yi Lu and the coauthors

**Response to Reviewer 1**

1. What is typhoon disaster? Many losses along the coastal countries are caused by storm surges and waves. So, how representative do rainfall and wind are?

We do appreciate the comment.

(1) For the first question, the typhoon disasters in this paper refer to affected population or direct economic losses caused by typhoons in Zhejiang province. In order to clarify the criteria for typhoon disaster, we have added following definition in the beginning of the abstract.

"a study on risk zoning of typhoon disasters (typhoon disasters in this paper refer to affected population or direct economic losses caused by typhoons in Zhejiang province) is carried out."

(2) For the second question, we explain this from following three aspects.

A. In this paper, we agree that typhoon disasters usually cause three disasters: heavy rain, high winds and storm surges, which has been pointed out at the beginning of "Introduction".

B. The reasons for choosing typhoon rain and typhoon wind are mainly based on following two considerations: 1) the typhoon storm surge disaster in Zhejiang province is relatively small compared to the other two disasters; 2) considering the consistency of coastal and inland areas in the study, we finally choose considering the effects of typhoon rain and typhoon wind. Therefore, the typhoon rain and typhoon wind in this paper can basically represent the factors causing typhoon disasters.

C. This is the limitation of the study, and we will explore the role of storm surges in future work.

(3) To explain this clearly in this paper, we have added the following explanation at the end of section 1:

"This paper does not consider the impact of storm surges. The factors causing typhoon disasters are represented by typhoon rain and typhoon wind."

And the following discussions have been added at the end of section 5 "Discussion and Conclusions".

"This paper mainly considers the effects of typhoon rain and typhoon wind, without considering the impact of storm surge. This is the limitation of the study, and we will explore the role of storm surges in future work."

2. "Risk of Typhoon Rainstorms" shall be "Probability of Typhoon Rainstorms". So does "Risk of Typhoon Winds".

Thanks so much for the suggestion. According to the suggestion, "Risk of Typhoon Rainstorms"

all have been changed into "Probability of Typhoon Rainstorms". So does "Risk of Typhoon Winds".

3. Risk Assessment (by using hazards, i.e., rainfall, wind and social vulnerability) results shall be validated with historical losses. How good/valid/bad/invalid is the assessment?

Thanks for your suggestion. In this paper, typhoon disaster comprehensive risk index is obtained by combining the factors causing typhoon disasters and vulnerability, which is a dimensionless result. Based on the comprehensive risk index, five risk grades for typhoon disasters are defined.

To illustrate the quality of the assessment, the distribution of typhoon disaster losses in Zhejiang province from 2004 to 2012 are given at the beginning of section 3.

Comparing the high and low value areas of Figure 3 and Figure 11, we find that the index in Figure 11 presents a good reflection of the distribution of typhoon disasters in Zhejiang province (Figure 3), especially in the southeastern coastal areas. The southeast coastal areas face the highest risk, especially in the boundary regions between Zhejiang and Fujian province, and Taizhou and Wenzhou cities. Overall, the risk of typhoon disasters decreases from the coast to inland areas.

We have given a conclusion at the beginning of the last paragraph in Section 4.3 to clarify that the evaluation results are good.

4. Social vulnerability has always been changing along time. The authors developed the index based on methods of past studies. The question is, how representative of the data (of one year, of several years) could be? Explanation shall be given. (btw, simplified social vulnerability as an index can be useful, but it could also be highly misleading for many risk management scenarios.)

Thanks for your thoughtful suggestion.

The population data used in this paper is obtained from the sixth national population census of the Population Census Office of the National Bureau of Statistics of China and the 2010 statistical yearbooks of each city in Zhejiang province published by the cities' statistical bureaus. The census data is updated every six years, and the 2010 census results are exactly during 2004-2012 which is the research period. Therefore, the population data for 2010 in this paper can basically represent the population vulnerability of this period. In addition, there exist many missing and abnormal records in the original data, which take a long time to be processed. This article focuses on typhoon disaster risk zoning in Zhejiang province, so we didn't discuss the difference of population vulnerability between 2010 to early year. The variation of population vulnerability is an interesting topic. Maybe we can discuss it in future work.

To clarify the representative of the population data used in this study, we have added following sentences in section 2.1.2 "Disaster and Social Data".

"The census data is updated every six years, and the 2010 census results are exactly during 2004-2012 which is the research period. Therefore, the population data for 2010 in this paper can basically represent the population vulnerability of this period.".

5. In this paper, there are quite some mis-used/well-defined terms (risk).

Thanks very much for the suggestion.

According to the suggestion, "Risk of Typhoon Rainstorms" all have been changed into "Probability of Typhoon Rainstorms". So does "Risk of Typhoon Winds", and the corresponding conclusions and abstracts have been revised. The specific amendments are as follows.

"3.1 Risk of Typhoon Rainstorms" have been changed into "3.1 Probability of Typhoon
Rainstorms".
"3.2 Risk of Typhoon Winds" have been changed into "3.2 Probability of Typhoon Winds".
The last sentence of section 3.1 have been modified as "There are three centers of high probability:
Taizhou, Wenzhou and Ningbo cities.".
The last two sentences of section 3.2 have been modified as "The areas at high probability of
typhoon winds are consistent with those with a high probability of typhoon rain, i.e. Wenzhou,
Taizhou and Ningbo cities. The probability of typhoon extreme winds is much higher in coastal
areas than inland.".

**Response to Reviewer 3**

General Comments:
1. Manuscript required a lot of editing for proper English language usage. Some of this reviewer's
recommended changes may have changed the intent of the authors. If that is the case, then the
sentence or paragraph should be rewritten.
We do appreciate the comment. Based on your suggestions, we have rewritten or revised the paper
one by one. Details are shown in answers to "Specific Comments".
2. The study mentions storm surge as one of the three factors affecting "typhoon" disasters, yet
there is no assessment of the effects of storm surge. Storm surge kills many more people than wind.
If wind intensity is used as a proxy for storm surge, that should be indicated. Tropical cyclone size
is an important consideration for storm surge and is sometimes more important than intensity. This
should be mentioned as a limitation of the study.
Thanks very much for the comments and suggestions. We explain this from following three
aspects.
A. In this paper, we agree that typhoon disasters usually cause three disasters: heavy rain, high
winds and storm surges, which has been pointed out at the beginning of "Introduction".
B. The reasons for choosing typhoon rain and typhoon wind are mainly based on following two
considerations: 1) the typhoon storm surge disaster in Zhejiang province is relatively small
compared to the other two disasters; 2) considering the consistency of coastal and inland areas in
the study, we finally choose considering the effects of typhoon rain and typhoon wind. Therefore,
the typhoon rain and typhoon wind in this paper can basically represent the factors causing
typhoon disasters.
C. This is the limitation of the study, and we will explore the role of storm surges in future work.
In addition, To explain this clearly in this paper, we have added the following explanation in
original section 1:
"This paper does not consider the impact of storm surges. The factors causing typhoon disasters
are represented by typhoon rain and typhoon wind."
And the following discussions have been added at the end of section 5 "Discussion and
Conclusions".
"This paper mainly considers the effects of typhoon rain and typhoon wind, without considering
the impact of storm surge. This is the limitation of the study, and we will explore the role of storm surges in future work."

3. The first line of the Introduction equates "typhoon" to "tropical cyclone". The term "tropical cyclone" generally includes tropical depressions, tropical storms, and typhoons. This should be clarified at the beginning.

Thanks for your comment. According to your suggestion, we have clarified "typhoon" at the beginning of "Abstract". Following sentence has been added.

" In this paper, typhoon simply means tropical cyclone.".

4. How do you define a disaster? What are the criteria?

Thanks for your question. The typhoon disasters in this paper refer to affected population or direct economic losses caused by typhoons in Zhejiang province. In order to clarify the criteria for typhoon disaster, we have added following definition in the first sentence of the abstract.

"a study on risk zoning of typhoon disasters (typhoon disasters in this paper refer to affected population or direct economic losses caused by typhoons in Zhejiang province) is carried out."

5. For consistency, use either "indices" or "indexes".

Thanks for your comment. According to your suggestion, we use "indexes" uniformly in this paper.

Specific Comments:

1. Abstract:

a. Lines 463-464: Doesn't make sense: Do you mean "Assuming that risk signifies probability of hazard events and that future probability is the same as historical probability for a specific period, we can understand risk by learning from past events." If not, please rewrite the sentence.

b. Line 465: "…over mainland China during 1980-2014 and disaster and …"

c. Line 470: "The above analyses…"

d. Line 474: Locate Hangzhou Bay on your map.

e. Lines 475-477: Move this sentence to the end of the paragraph (after line 482).

Reply:

a. Your understanding is what we means. Supposing future probability is the same as historical probability for a specific period, we can understand risk by learning from past events.

Suggestion b and c have been applied in this paper.

d. We have located Hangzhou Bay on the map according to the suggestion.

e. Considering that we have analyzed population vulnerability before discussing typhoon disaster comprehensive risk of Zhejiang province, we tend not to move this sentence to the end of the paragraph.

2. Introduction:

a. Lines 486-487: See Iten 3 under General Comments.

b. Line 490: Delete "the".

c. Line 497: Delete "had"

d. Line 504: I believe that this reference is for R. A. Pielke, Jr. (not his father Sr.). Check this out here and at lines 857-860 in the reference section.

e. Lines 521: Consider making two separate sentences: "…disaster loses. Few studies…"

Reply:

a. According to your suggestion, we have clarified "typhoon" at the beginning of "Abstract".

Amendments have been made according to the suggestion b c and e.

d. Line 504: we have checked this out here and revised Lines 857-860 in the reference section as follows.

"Pielke, R. A. J. and Landsea, C. W.: Normalized hurricane damages in the United States: 1925-95,

Weather & Forecasting, 13(3), 621--631, 1998.

Pielke, R. A. J., Gratz, J., and Landsea, C. W.: Normalized hurricane damage in the United States:

1900–2005, Natural Hazards Review, 9(1), 29-42, 2008.".

3. Data and Methods:

a. Line 531: Do you mean "north of" instead of "south of"?

b. Line 534 and Figure 1: Identify Hangzhou Bay on the map and add to Legend. The legend is not as clear as stating "(in red)" after 'major cities" and add "in provinces (in black)". Also, does the scale need to be in kilometers as well as in miles?

c. Line 540: "…from the National…"

d. Line 544: ";;;from 2004 to 2012. So when calculating the intensity index of factors causing typhoon disasters, the time…"

e. Lines 547-549: Do you mean: "…(see details in section 3.1 and 3.2), we assume that future probability is the same…"? Otherwise, please rewrite.

f. Line 551: "…period for both. In addition, the Objective Synoptic Analysis Technique (OSAT)

method…"

g. Line 552: "…details…"

h. Line 559: "…being 17 at XXXX" ; add either Wenzhou or Taizhou. Also, se General comments

Item 4.

i. Where is "Quzhou"; indicate on the map.

j. Figure 2, line 566: See comment in General Comments Line 4.

k. Line 572: "…structural analysis of a precipitation…

l. Line 584: "…combinations…"

m. Line 595: "…calculating the intensity…"

n. Line 597: "…calculating the typhoon…"

o. Lines 592-599: Reference "MIN-MAX standardization" and "Z-score standardization"

techniques' Explain rationale for selecting the Z-score technique as you did for the MAXMIN

technique.

p. Lines 607-608: Consider: "Based on various SoVIs derived for disaster social vulnerability in

America, Chen et al. (2014) selected 29 variables as proxies…"

q. Line 609: Consider: "…. We then use these vulnerability indexes to calculate the population…"

Reply:

a. Yes, it should be "north", we have modified it.

b. According to the suggestions, we have redrawn Figure 1.

Suggestion c d f g h k l m n j p and q have been applied in this paper.

e. Yes, your understanding is what we mean.

h. To clarify the criteria for typhoon disaster, we have added following definition in the first sentence of the abstract.

"a study on risk zoning of typhoon disasters (typhoon disasters in this paper refer to affected population or direct economic losses caused by typhoons in Zhejiang province) is carried out."

i. "Quzhou" have been indicated on the original map, please check.

o. To explain rationale for selecting the Z-score technique, we have rewrote sentence "The Z-score standardized method is based on the mean and standard deviation of the raw data.".

"The Z-score standardized method is based on the mean and standard deviation of the raw data, which is prepared for CCA method.".

4. Typhoon Disaster Losses and Factors:

a. Page 19: Discuss Figure 3c and 3d in the text. Do not need the term "unit" in the caption.

b. Line 630: Locate "Quzhou city" on the map.

c. Line 632: "…typhoon disasters, "typhoon rainstorm" means…and "torrential rainstorm" means…"

d. Lines 652-661: I assume that "grades" pertains to some kind of typhoon wind damage scale. Is that correct. In any event, you need to define wind values for Grades 6, 8, 10, and 12.

e. Line 661: "…and are only…"

f. Line 663: Do you mean: "…with those with a high risk of typhoon…"

g. Line 667: Rewrite: "…typhoon winds of (a) grade 6 or above, (b) grade 8 or above, (c) grade 10 or above, and (d) grade 12 or above."

Reply:

a. According to the suggestions, we have added following discuss of Figure 3c and 3d in the text.

"According to the statistical yearbooks of each city in Zhejiang province, Jiaxing, Shaoxing, Hangzhou in the northeast, and Wenzhou, Jinhua and Taizhou in the southwest are the regions with the largest agricultural planting area, with more agricultural population in the southwest. Only parts of the plain area were affected by serious agricultural disasters in the northeast. The agricultural disaster areas in the southwest are wider. (Fig. 3c). According to the main indicators of Zhejiang's national economy (total GDP and per capita GDP), the central cities such as Hangzhou in the northeast had the most developed economy, and the urban economies of Wenzhou and Taizhou in the southwest were also relatively good. However, the economic losses in southwestern Zhejiang are severe, much higher than in the northeastern cities. (Fig. 3d).".

b. "Quzhou" have been indicated on the original map, please check.

c. To clarify the criteria for typhoon disaster, we have added a definition in the first sentence of the abstract.

d. and g. According to national standards of China for tropical cyclone ratings (GBT 19201-2006), tropical cyclones are classified into six grades according to the maximum wind speed on the ground near the center, including Grade 6-7, Grade 8-9, Grade10-11, Grade 12-13, Grade 14-15 and Grade 16 or above. In this paper, we give 4 grades to describe typhoon winds in Zhejiang province. To clarify these grads, we have added definitions in Figure 7 as follows.

"Figure 7. Probability of the occurrence of typhoon winds in Zhejiang province at (a) grade 6 or above ($\geq$ 10.8 m/s), 672 (b) grade 8 or above ($\geq$ 24.5 m/s), (c) grade 10 or above ($\geq$ 32.7 m/s) and (d) grade 12 or above ($\geq$ 41.5 m/s) ".

Suggestion e and f have been applied in this paper.

5. Risk Assessment and Regionalization of Typhoon Disasters
a. Line 676: Are factors and hazards used interchangeably? Storm surge is mentioned again as a
"main factor", yet there is not storm surge losses or assessments in then study.
b. Line 680: "…includes…"
c. Table 1, line 690, first row of Table: Spelling/typo: "Disasters"
d. Lines 699-700: "…and Ningbo cities are identified, which is consistent with the nresults of
Chen et al. (2011).…"
e. Line 708, Table 2: Define how "primary", "secondary' and "tertiary" industries differ.
f. Lines 744 and 766, Figures 10 and 11: Why is the island northeast of Ningbo city of "low risk",
while Ningbo is medium to high risk?
g. Lines 777-778: "…and a better educated population."
h. Line 787: "…further study…."
Reply:
a. Yes, factors and hazards here can used interchangeably. However, we use "factors" uniformly in
this part, which represent typhoon precipitation and typhoon wind causing disasters.
For the second question, this paper mainly considers the effects of typhoon precipitation and
typhoon wind, without considering the impact of storm surge, which have been answered in
General Comments 2. To make it clear, we modified this sentence as follows.
"The main factors causing typhoon disasters, which are considered in this study, are rainstorms
and winds.".
Suggestion b c d and g h have been applied in this paper.
e. The definitions of "primary", "secondary' and "tertiary" industries are formulated in accordance
with the National Economic Industry Classification (GB/T4754-2002). This is a consensus, so it is
not specified in the Table 2. The primary industry refers to agriculture, forestry, animal husbandry
and fishery. The secondary industry refers to the mining, manufacturing, electricity, gas and water
production and supply industries, and the construction industry. The tertiary industry refers to
other industries except the primary and secondary industries.
f. The island northeast of Ningbo city is Zhoushan city. Typhoon disaster comprehensive risk
index is obtained by combining the factors causing typhoon disasters and population vulnerability,
and these two indexes in Ningbo city are higher than in Zhoushan. According to the statistical
yearbooks of each city in Zhejiang province, the population density of Ningbo City is much larger
than that of Zhoushan, where population was more vulnerable. In addition, Ningbo city was more
affected by typhoon and precipitation and typhoon wind in 2004-2012. So Zhoushan city is low
risk, while Ningbo is medium to high risk.
6 References:
a. Lines 798, 846, and 888: Capitalize: "China".
b. Line 859: Capitalize "United States"
c. Lines 857-860 Check Pielke Jr vice Pielke (the father).
Reply:
Suggestions have been applied in this paper.

[revised manuscript text omitted]